# Vitamin B12 Levels, Substance Use Patterns and Clinical Characteristics among People with Severe Substance Use Disorders: A Cohort Study from Western Norway

**DOI:** 10.3390/nu14091941

**Published:** 2022-05-05

**Authors:** Tesfaye Madebo, Mitra Bemanian, Jørn Henrik Vold, Ranadip Chowdhury, Christer Frode Aas, Karl Trygve Druckrey-Fiskaaen, Kjell Arne Johansson, Lars Thore Fadnes

**Affiliations:** 1Department of Pulmonary Medicine, Stavanger University Hospital, P.O. Box 8100, 4068 Stavanger, Norway; 2Bergen Addiction Research, Department of Addiction Medicine, Haukeland University Hospital, Østre Murallmenningen 7, 5012 Bergen, Norway; mitra.bemanian@student.uib.no (M.B.); jorn.henrik.vold@helse-bergen.no (J.H.V.); christer.frode.aas@helse-bergen.no (C.F.A.); karl.trygve.druckrey-fiskaaen@helse-bergen.no (K.T.D.-F.); kjell.johansson@uib.no (K.A.J.); 3Department of Global Public Health and Primary Care, Faculty of Medicine, University of Bergen, P.O. Box, 7804 Bergen, Norway; 4Centre for Health Research and Development, Society for Applied Studies, 45, Vijay Mandal Enclave, Kalu Sarai New Delhi, Delhi 110016, India; ranadip.chowdhury@sas.org.in

**Keywords:** vitamin B12, substance use disorder, micronutrients deficiencies, liver disease, opioid agonist therapy, Norway

## Abstract

People with severe substance use disorder (SUD) have a higher burden of micronutrient deficiency compared with the general population. The aim of this study was to investigate vitamin B12 status and risk factors of deficiency related to substance use, opioid agonist therapy (OAT), as well as hepatitis C infection and liver fibrosis. In this prospective cohort study, participants were recruited from outpatient OAT and SUD clinics in western Norway, and assessed annually with a clinical interview and exam, including venous blood sampling. Data were collected between March 2016 and June 2020, and a total of 2451 serum vitamin B12 measurements from 672 participants were included. The median serum vitamin B12 concentration was 396 (standard deviation 198) pmol/L at baseline, 22% of the population had suboptimal levels (<300 pmol/L) and 1.2% were deficient at baseline (<175 pmol/L). No clear associations were seen with substance use patterns, but liver disease and younger age were associated with higher vitamin B12 levels. Although the majority of participants had satisfactory vitamin B12 levels, about a fifth had suboptimal levels that might or might not be adequate for metabolic needs. Future studies could investigate potential gains in interventions among patients with suboptimal but non-deficient levels.

## 1. Introduction

Severe substance use disorder (SUD) is associated with a multitude of comorbidities and complications, one of which is malnutrition [1]. This is, in part, attributable to the adverse physiological manifestations of chronic substance use on gastrointestinal, renal and hepatic functions [2]. However, poverty and limited food availability, as well as adverse dietary habits, are also important factors contributing to the poor nutritional status of people with severe SUD [1,3,4]. The diet of people with severe substance use disorders (SUD) is often skewed towards simple carbohydrates-added sugar, but in some cases also processed meat products, and is often lacking in important micronutrients [4,5], which in turn, puts them at high risk of developing many micronutrient deficiencies [4,6]. There is little literature on the prevalence of micronutrient deficiencies among people with severe SUD, and little known specifically about the status of vitamin B12 and risk factors of deficiency among people with severe SUD.

Vitamin B12 is an essential nutrient found primarily in foods of ruminant origin, including meat and dairy products [7,8]. It is an important co-factor in one-carbon methyl transfer reactions and has a vital role in cellular proliferation and genomic integrity [8]. Vitamin B12 deficiency is a known cause of megaloblastic anemia and neuronal demyelination [8,9]. This, in turn, makes vitamin B12 deficiency a risk factor for a vast array of neurological and neuropsychiatric disorders [8,9]. Since the body has a substantial storage capacity of vitamin B12 in the liver, deficiency and related complications usually develop gradually over years of imbalanced vitamin B12 homeostasis. Liver disease, including hepatitis C infection and alcoholic liver disease, elevates the concentration of circulating B12 by depleting tissue storage [10,11,12]. These conditions are highly prevalent among people with severe SUD, potentially complicating the assessment of vitamin B12 status in these populations [13].

People with severe SUD have a markedly high disease burden compared with the general population, including micronutrient deficiencies. There are limited data available about the status of vitamin B12 among people with severe SUD, and it would be helpful to identify potentially reversible factors to reduce their disease burden. We aimed to assess the vitamin B12 status and the prevalence of deficiency among people with severe SUD from outpatient opioid agonist therapy (OAT) clinics and municipal health care clinics in Bergen and Stavanger, Norway. Furthermore, we aimed to measure the effect of substance use, OAT, hepatitis C infection, and liver fibrosis on suboptimal and deficient serum vitamin B12 concentration.

## 2. Materials and Methods

### 2.1. Study Characteristics; Design, Population, Data Collection and Study Sample

This is a prospective cohort study presenting data drawn from the multicenter studies INTRO-HCV [14] and ATLAS4LAR [15]. Participants were recruited from an outpatient population visiting OAT clinics or municipal health care clinics for severe SUD in Bergen and Stavanger, Norway. Participants were assessed annually with a research nurse-led and questionnaire-based interview focused on somatic and mental health, psychosocial aspects and substance use patterns. Data were collected using the software CheckWare. Clinical data were obtained from the electronic medical record. Data collected between March 2016 and June 2020 are presented. A total of 2451 serum vitamin B12 measurements from 672 participants were included.

### 2.2. Measuring Serum Vitamin B12; Laboratory Assays and Definitions

Venous blood samples were collected and analyzed at the Department of Medical Biochemistry and Pharmacology at Haukeland University Hospital in Bergen and the Department of Medical Biochemistry at Stavanger University Hospital in Stavanger (both accredited by ISO-standard 15189) for analysis of vitamin B12 concentration. The department in Bergen used the Electrochemiluminescence Immunoassay (7% analytical variation at 314 pmol/L), and the laboratory in Stavanger used the Chemiluminescense Microparticle assay (7% analytical variation at 300 pmol/L). Data on serum vitamin B12 concentration were obtained from the electronic medical record. The unit used was picomoles per liter (pmol/L). Two separate cut-offs were used when describing vitamin B12 status: s-vitamin B12 < 175 pmol/L [16,17,18] as vitamin B12 deficient, and s-vitamin B12 < 300 nmol/L [19] as suboptimal.

### 2.3. Study Variables, Baseline, Clinical and Sociodemographic Factors

Baseline was defined as the serum vitamin B12 measurement closest to the first annual health assessment for each individual. Subsequent serum vitamin B12 measurements were listed chronologically and included as follow-up, and time was defined as years from the baseline. OAT was defined as receiving methadone or buprenorphine-based medication. We calculated an OAT dose ratio corresponding to the prescribed daily dose of medication divided by the expected mean dose (90 mg for methadone and 18 mg for buprenorphine) according to WHO [20]. As for the clinical factors, injecting substances was defined as having injected any substance within the prior six months. Frequent substance use was defined as using any of the following substances on a minimum weekly basis during the 12 months leading up to the annual health assessment: alcohol, cannabis, benzodiazepines, stimulants (amphetamines and cocaine) and non-prescribed opioids (e.g., heroine). Hepatitis C status was determined by means of a quantitative polymerase chain reaction assay, and diagnosed HCV RNA was defined as hepatitis C infection. The risk of hepatic fibrosis was estimated using the Fibrosis-4 (Fib-4) scoring system [21], defining Fib-4 < 1.45 as advanced fibrosis to be unlikely and Fib-4 > 3.25 as likely advanced fibrosis. Regarding sociodemographic factors, housing conditions for the last 30 days leading up to the annual health assessment were defined as stable (living in owned or rented home or at an institution) or *unstable* (being homeless, living at a shelter, or with friends and family). Age was categorized into the following groups: <30 years, 30–39 years, 40–49 years, 50–59 years and ≥60 years. The electronic medical records of participants with suboptimal and low levels of vitamin B12 (<300 pmol/L) were examined for the presence of chronic diseases such as chronic gastritis, Crohn’s diseases and ulcerative colitis, Helicobacter pylori infection, chronic viral hepatitis (hepatitis B or C), gastrointestinal surgery such as gastric bypass or removal of the terminal ileum, alcohol-related disease and alcohol dependence, cancers, use of protein pump inhibitors or metformin, and presence of documented low folate level.

## 3. Statistical Analyses

Stata/SE 16.0 (Stata Corporation) was used for the generation of descriptive data and a linear mixed model. SPSS version 26.0 (IBM) was used for expectation maximization imputation. The software R version 4.0.3 (R foundation for Statistical Computing) with the package *mgcv* was used for the preparation of generalized additive models and their graphical presentations. The website Sankeymatic (sankeymatic.com/build, accessed 01 October 2021) was used for the generation of a Sankey diagram. The threshold of statistical significance was set to *p* < 0.05 for all analyses. Nine percent of values were missing across the sociodemographic and clinical variables. These were assumed to be “missing at random” and expectation maximization imputation was performed to replace them with estimated values [22]. Descriptive data is presented with total numbers and percentages. Median serum vitamin B12 concentration of the cohort is presented with interquartile range (IQR). The prevalence of suboptimal and deficiency of vitamin B12 is presented as percentages with 95% confidence intervals. Generalized additive models and plots were generated to visualize the nonlinear associations of substance use severity with serum vitamin B12 concentration, adjusted for gender and age. To do this, a substance use severity index was generated based on the type, frequency and number of substances used (for details see Appendix A) [23]. A linear mixed model regression was performed in order to estimate associations of clinical and sociodemographic factors with serum vitamin B12 concentration at baseline, as well as the time interactions of these associations. The model was random intercept fixed slope with the estimator set to restricted maximum likelihood. Time was defined as years from baseline, and all predictor variables were kept constant to values held at baseline. Time interactions were added to investigate the impact of time on associations between predictors with serum vitamin B12 concentration.

## 4. Results

The mean age at baseline was 44 years (SD: 11), and 71% were male (Table 1). Eighty-eight percent (581/663) were patients enrolled in OAT, and of these 61% were prescribed buprenorphine-based medications and 39% were prescribed methadone. Twelve percent lived under unstable housing conditions, and 53% had injected substances during the last six months leading up to the baseline assessment. Fifty-three percent of the cohort were infected with HCV and 32% had Fib-4 scores indicating advanced liver fibrosis. Weekly substance use was reported by 76%, and the most common substances used on a weekly basis were cannabis (49%) and benzodiazepines (38%), followed by stimulants (26%) and alcohol (25%).

The median (IQR) serum vitamin B12 concentration of the study cohort was 396 (198) pmol/L at baseline. Twenty-two percent (CI: 19–25) of the population had low serum vitamin B12 levels (<300 pmol/L) and 1.2% (CI: 0.6–2.3) were deficient at baseline (<175 pmol/L). The median (IQR) mean corpuscular volume (MCV) for those with sufficient, suboptimal and deficient vitamin B12 levels at baseline were 91 (87–95), 91 (88–94) and 95 (89–99), respectively (Appendix A). A large majority (81%) of the participants maintained their baseline vitamin B12 status in the subsequent assessment (Figure 1).

The linear mixed model regression showed that those in the older age groups had lower serum vitamin B12 concentrations at baseline compared with those in younger age categories. The mean difference for 50–59 years compared with those under the age of 30 years was −49 pmol/L (CI: −94, −4.7), and for those above the age of 60 years (−91 pmol/L, CI: −153, −29) (Table 2).

Higher scores on Fib-4 were associated with higher serum vitamin B12 concentration at baseline (29 pmol/L, CI: 20, 38), however, with a near-significant negative time trend (−3.7, CI: −8.3, 0.9). No other sociodemographic or clinical variables were associated with serum vitamin B12 concentration at baseline, and none of the time trends were significant. The generalized additive models showed no clear associations between substance use severity and vitamin B12 concentration in serum (Figure 2).

Lower levels of serum vitamin B12 (<300 pmol/L) were not associated with known risk factors such as presence of chronic gastritis, Crohn’s diseases and ulcerative colitis, gastrointestinal surgery such as gastric bypass or removal of the terminal ileum, Helicobacter pylori infection, chronic viral hepatitis (hepatitis B or C), alcohol-related disease and alcohol dependence, cancers, use of protein pump inhibitors, or metformin.

## 5. Discussion

The serum concentration of vitamin B12 in this population of outpatients with severe SUD was predominantly within normal range, whereas only 1.2% have clear deficient vitamin B12 levels (<175 pmol/L). A fifth had levels that might have been suboptimal with vitamin B12 level < 300 pmol/L). A large majority of the participants maintained their baseline vitamin B12 status in the subsequent assessments. Of all the sociodemographic and clinical variables included, only older age and the hepatitis virus infection-related liver fibrosis estimate Fib-4 were associated with serum vitamin B12 concentration. Being in the oldest age groups, i.e., >50 years, was associated with lower serum vitamin B12 concentration, a finding that is in line with existing literature [24]. This might have been linked to a higher prevalence of undiagnosed pernicious anemia among older people, which could be linked to reduced absorption of vitamin B12. Substance use patterns were, for the most part, not significantly associated with vitamin B12 levels, although there was a near-significant trend between frequent use of stimulants and lower vitamin B12 concentration. In the partly adjusted linear mixed model, the use of stimulants on a minimum weekly basis was associated with lower vitamin B12 concentration at baseline compared with less frequent or no use—however, this association was non-significant in the model that included all variables. Previous studies have found lower vitamin B12 levels among OAT patients reporting using stimulants upon admission to treatment [25], and among methamphetamine users compared with healthy controls [26].

Our study showed higher levels of vitamin B12 among individuals with liver disease, suggesting that liver disease plays a role in the turnover of vitamin B12. Similar findings have been described in studies on people with alcoholic-related liver disease and chronic hepatitis C infection [10,11,12], adding to the evidence that fibrotic/inflammatory liver disease raises the concentration of circulating vitamin B12 by depleting hepatic tissue stores. Other studies have found correlations between vitamin B12 concentrations in blood with hepatic enzymes, suggesting that the concentration of circulating vitamin B12 could be related to hepatic disease severity [27,28,29]. In line with this, a retrospective study on patients with chronic hepatitis C infection undergoing antiviral treatment found that lower vitamin B12 levels at baseline were associated with better outcomes and higher sustained viral response [12].

In our subsample of participants with suboptimal or deficient levels, we did not identify any known risk factors of vitamin B12 deficiency such as inflammatory bowel diseases or having gone through gastrointestinal surgery. Neither substance patterns were associated with lower levels. This could indicate that this group had a lower intake of vitamin B12-rich foods (fish, dairy, egg, meats) and did not use vitamin B12 containing supplements. Nevertheless, vitamin B12 deficiency was much lower (1.2%) than reported from selected large surveys from countries in North America and Europe which were about 2–10% with much higher prevalence in the older age group [8]. Some of the reasons for differences in these studies may be related to the use of various methods to measure vitamin B12, different cut-off values, unit of report, reference range and study population [30]. However, prevalence of vitamin B12 deficiencies is lower than in several other studies, but another study from Norway also indicates that intake of animal products including meat and dairy is relatively high [4].

The main weakness of our study related to our interpretations of Vitamin B12 level was the absence of measurement of halotranscobalamin, homocysteine and methylmalonic acid to aid the interpretation of vitamin B12 levels. However, we used mean corpuscular volume to fill some of this gap. We also have limited clinical information about neurological manifestations that may occur as hematological complications of deficient vitamin B12 levels [31]. Patients tend to present with either hematologic or neurologic symptoms in many cases [32]. The median value of mean corpuscular volume was similar among those with normal, suboptimal and deficient levels of vitamin B12, with an increase in size among those with deficient levels as a response to aging erythrocytes with deficient levels of vitamin B12. Systematic review showed that mean MCV can be within the normal range with vitamin B12 deficiency [33]. The mean corpuscular volume generally tends to increase before the manifestation of anemia, but the presence of megaloblastic anemia alone is not a reliable way to diagnose B12 deficiency, as B12 deficiency has to be relatively severe before anemia appears [34].

Most of total serum vitamin B12 is haptocorrin-bound and thus not fully reflective of cellular vitamin B12 status. Functional vitamin B12 deficiency can therefore be present even when serum B12 concentrations are normal [35]. Several studies have suggested that transcobalamin (the metabolically active portion) is a better index of vitamin B12 deficiency but is not yet used widely because of cost and limited availability [36]. Aiding interpretation with methylmalonic acid or homocysteine as the second-line test could have increased the precision of vitamin B12 deficiency, particularly within the mid-group of suboptimal levels [37]. The amount of people with suboptimal vitamin B12 levels of about 22% probably includes both a majority of people with adequate and a minority with inadequate levels for metabolic requirements.

In our study, about 22% have suboptimal vitamin B12 levels but others had reported higher prevalence (30–60%) from different settings including the western population, in pregnant women, and in less-developed countries [38,39]. The benefits of treating suboptimal vitamin B12 are unknown because of a lack of good quality clinical trial data. However, some studies have suggested treating with very low oral or short courses of oral vitamin B12 [7] but the strength of recommendation is weak and the quality of evidence is low [37]. Others recommend that patients with persistently suboptimal vitamin B12 levels should be investigated for anti-IF antibody titers and additional biochemical investigations to confirm biochemical deficiency [40]. There is a need for detailed, clinically focused studies for suboptimal vitamin B12 level in future studies.

However, our study also has strengths, including its prospective cohort design over a 4-year period presenting data from several centers within western Norway with a relatively large population of people who are generally difficult to reach. Our data emphasize the importance of increasing general awareness and replacement treatments in at-risk populations.

## 6. Conclusions

Most patients have a satisfactory serum level of vitamin B12 and have maintained their baseline vitamin B12 status in the subsequent assessments. Only 1.2% have vitamin B12 deficiency, but about a fifth of the patients have suboptimal levels that might or might not be adequate for metabolic needs. No clear associations were seen with substance use patterns, but liver disease and younger age were associated with higher vitamin B12 levels. Future studies could investigate potential gains in interventions among patients with suboptimal but non-deficient levels.

## Figures and Tables

**Figure 1 nutrients-14-01941-f001:**
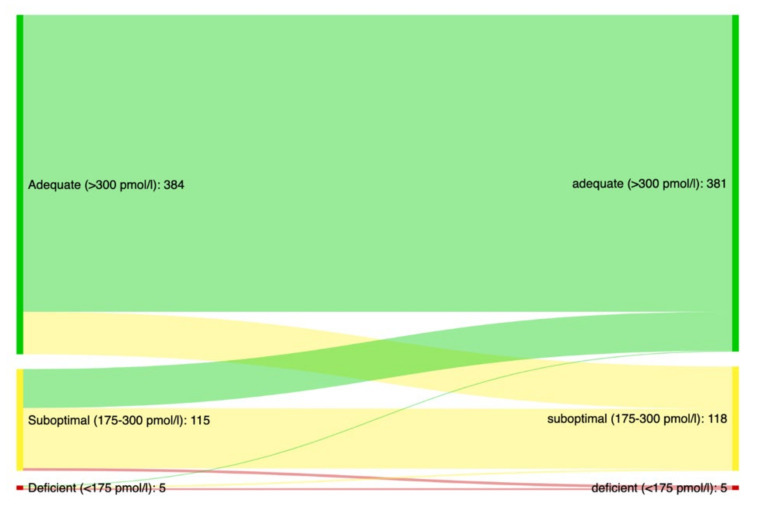
Distribution of vitamin B12 status in the first and second annual assessments. This figure displays the changes in vitamin B12 status categories from the first (left) to the second (right) assessment for participants with at least two vitamin B12 measurements.

**Figure 2 nutrients-14-01941-f002:**
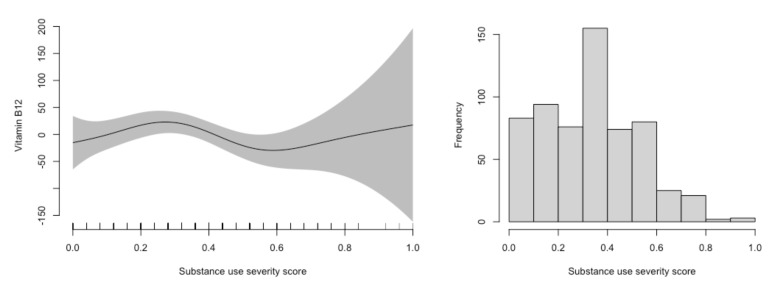
Association of substance use severity with serum vitamin B12 concentration. The figure to the left was constructed using generalized additive models in R and displays of the associations of vitamin B12 concentration in serum with the substance use severity index. The solid line depicts the association at various severity scores, whereas the shaded area represents the 95% confidence interval of this association. The figure to the right displays the distribution of substance use severity scores in the population.

**Table 1 nutrients-14-01941-t001:** Baseline characteristics of the cohort.

Characteristic	N (%)
**Gender**	
Male	474 (71)
Female	198 (29)
**Age group**	
<30 years	78 (12)
30–39 years	192 (29)
40–49 years	204 (30)
50–59 years	157 (23)
≥60 years	41 (6)
**Education level**	
Not completed primary school	39 (6)
Primary school (9 years)	300 (45)
High school (12 years)	269 (40)
≤3 years higher education	51 (8)
>3 years higher education	13 (2)
**Housing condition ^1^**	
Unstable	81 (12)
Stable	591 (88)
**HCV infection ^2^**	315 (53)
**Fib-4**	
<1.45	452 (68)
>3.25	40 (6)
**Injecting substances ^3^**	325 (53)
**Opioid agonist therapy**	
Buprenorphine	352 (53)
Methadone	229 (35)
Not in OAT	82 (12)
**Weekly substance use ^4^**	
Alcohol	151 (25)
Cannabis	302 (49)
Stimulants ^5^	162 (26)
Benzodiazepines	233 (38)
Non-prescribed opioids	87 (14)
No weekly substance use	145 (24)
**Serum vitamin B12**	
Median pmol/L (IQR ^6^)	396 (198)
% with suboptimal levels (CI ^7^)	22 (19–25)
% with deficient levels (CI ^7^)	1.2 (0.6–2.3)

^1.^ Stable housing included living in owned or rented housing or at an institution, unstable housing included homelessness, living at temporary camping sites or with friends or family. ^2.^ Hepatitis C virus infection, defined as non-zero values on a quantitative HCV-RNA assay at baseline. ^3.^ Self-reported injection of any substance during the 6 months prior to the first health assessment. ^4.^ Self-reported substance use on a minimum weekly basis during the 12 months prior to the first health assessment. ^5.^ Amphetamine, methamphetamine or cocaine. ^6.^ IQR, interquartile range. ^7^ CI, 95% confidence interval.

**Table 2 nutrients-14-01941-t002:** Linear mixed model of serum vitamin B12 concentration (pmol/L) adjusted for sociodemographic and clinical factors, including substance use patterns.

Fixed Effects				
	Partly Adjusted ^1^		Adjusted	
	Effect Estimate	Time Trend (per Year)	Effect Estimate	Time Trend (per Year)
	Estimate (CI)	Slope (CI)	Estimate (CI)	Slope (CI)
**Vitamin B12**			386 (331, 440)	*28 (5.2, 50)*
**Gender**				
Male			0 (reference)	
Female			−7.3 (−37, 22)	
**Age**				
<30			0 (reference)	
30–39			1.7 (−37, 40)	
40–49			−38 (−80, 3.9)	
50–59			−49 (−94, −4.7)	
≥60			−91 (−153, −29)	
**Hepatic markers**				
HCV			20 (−7.4, 47)	
Fib-4	*31 (22, 40)*	*−4.1 (−8.6, 0.37)*	*29 (20, 38)*	−3.7 (−8.3, 0.9)
**OAT dose ratio ^2^**	30 (−1.1, 60)	−11 (−29, 5.8)	19 (−12, 50)	−13 (−31, 5.2)
**Injecting substances ^3^**	−23 (−54, 8.4)	−3.9 (−20, 12)	−12 (−46, 23)	−4.9 (−23, 13)
**Weekly substance use ^4^**				
Alcohol	21 (−14, 57)	−9.6 (−28, 8.4)	10 (−25, 45)	−9.4 (−27, 9.1)
Cannabis	27 (−3.4, 58)	−11 (27, 4.4)	25 (−6.6, 56)	−11 (−27, 6.0)
Non-OAT opioids	−2.9 (−47, 42)	−4.0 (−28, 20)	11 (−36, 57)	−4.3 (−30, 22)
Stimulants ^5^	*−36 (−72, −1.1)*	−0.7 (−20, 18)	−35 (−74, 3.7)	0.19 (−22, 21)
Benzodiazepines	26 (−5.6, 58)	−3.7 (−20, 12)	21 (−13, 55)	2.0 (−16, 20)

The table displays the results of a linear mixed model (restricted maximum likelihood regression) estimating associations of serum vitamin B12 concentration (pmol/L) with sociodemographic and clinical predictor variables at baseline (effect estimates), as well as the impact of predictors on changes in serum vitamin B12 concentrations over time (time trends per year). Significant results are shown in italics. Explanations: CI, 95% confidence interval. ^1.^ Adjusted for gender and age. ^2.^ The patients prescribed daily dose of opioid agonist divided by the WHO mean expected dose (90 mg for methadone, 18 mg for buprenorphine). In this variable, zero represents no prescribed OAT medication. ^3.^ Self-reported injection of any substance during the 6 months prior to the first health assessment. ^4.^ Self-reported use of a substance at a minimum weekly basis during the 12 months prior to the first assessment. ^5.^ Amphetamine, methamphetamine and cocaine.

## Data Availability

The data presented in this study are available on request from the corresponding author.

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
