# Peer review of "Vitamin B12 Levels, Substance Use Patterns and Clinical Characteristics among People with Severe Substance Use Disorders: A Cohort Study from Western Norway"

_nutrients, 2022, doi:10.3390/nu14091941_

Round 1

Reviewer 1 Report

People with severe substance use disorder (SUD) have a markedly high disease burden compared to the general population, including micronutrient deficiencies. There is limited data available about the status of vitamin B12 among people with severe SUD. This prospective study was aimed to assess the vitamin B12 status and risk factors of deficiency related to substance use, opioid agonist therapy (OAT), as well as hepatitis C infection and liver fibrosis. The prevalence of deficiency among people with severe SUD from outpatient opioid agonist therapy (OAT) clinics and municipal health care clinics in Bergen and Stavanger (Norway) was evaluated. A total of 2451 serum vitamin B12 measurements from 672 participants were included.

            Most patients had satisfactory serum vitamin B12 concentrations. The median serum vitamin B12 concentration was 396 pmol/l at baseline, and 22% of the study population had suboptimal levels (<300 pmol/l). Only 1.2 % had vitamin B12 deficiency at baseline (<175 pmol/l). No clear associations were found with substance use patterns, but liver disease and younger age were associated with higher vitamin B12 levels. Higher serum level of vitamin B12 among subjects with liver disease suggests that liver disease plays a role in the turnover of vitamin B12.

            It would be of interest to know whether the portion of holohaptocorrin is higher in patients with liver disease compared to those without liver damage. The higher vitamin B12 serum concentration in liver disease could possibly due to release of stored vitamin B12 in the liver. 

Reviewer 2 Report

This manuscript describes relationship between vitamin B12 levels and multiple factors among individuals with a diagnosis of substance abuse in Norway. Plasma B12 level data was collected from medical records at baseline as well as 1 and 2 year intervals. The underlying rationale appeared to be that these individuals might exhibit nutritional deficiencies, but no comparison group was included to establish this. Moreover, the finding of 22% with sub-optimal baseline B12 levels suggests otherwise. 

 The data analysis revealed very limited findings. There was an association between higher liver fibrosis scores and higher B12, and B12 levels were decreased in older subjects, which is well-documented. Indeed, neither of these observations are particularly informative. Perhaps there is a suggestion that the ablility of liver to take up B12 is compromisied with fibrosis, but this is hardly surprising. The time analyis shown in FIg. 1 and the relationship between substance abuse severity score and B12 levels in Fig. 2 are illustrative of the lack of significant findings. Otherwise the manuscript is clearly written. 
